Evaluation of shaping ability, apical transportation and centering ratio of T-Endo Must, WaveOne Gold, and Trunatomy in resin blocks

Karaca Sakallı Aybüke aybkekaraca94@gmail.com
Ekici Mügem Aslı
1 Faculty of Dentistry, Deparment of Endodontics, Gazi University Ankara , Ankara , Turkey
Pawar Ajinkya
Electronic publication date: 2024 Oct 14
Publication date: 2024
Volume: 12
Electronic Location ID: e18088
Received 2024 May 6; Accepted 2024 Aug 22
Copyright: ©2024 Karaca Sakallı and Ekici
Copyright year: 2024
Copyright holder: Karaca Sakallı and Ekici
License: This is an open access article distributed under the terms of the Creative Commons Attribution License, which permits unrestricted use, distribution, reproduction and adaptation in any medium and for any purpose provided that it is properly attributed. For attribution, the original author(s), title, publication source (PeerJ) and either DOI or URL of the article must be cited.
License URL: https://creativecommons.org/licenses/by/4.0/

Keywords: Dental alloys, Endodontics, Root canal preparation, Titanium nickelide

Funding: Gazi University Scientific Research Projects Coordination Unit (BAP) TDH-2022-7457 This study was supported by the Gazi University Scientific Research Projects Coordination Unit (BAP), which provided support with the project code TDH-2022-7457. The funders had no role in study design, data collection and analysis, decision to publish, or preparation of the manuscript.

==============================
Purpose

This study aimed to compare the shaping abilities of different nickel-titanium file systems.

Materials and Methods

Sixty-six j-shaped resin blocks were randomly divided into three groups (n = 22): Group T-Endo MUST (TE), Group WaveOne (W), Group TruNatomy (TR). After canal preparation, the amount of material removed from the canal, the centering ratio of the file systems, the direction and amount of canal transportation, and shaping errors were evaluated. Shaping time was calculated. Data of shaping time were analyzed with ANOVA and Tamhane test. Data on the shaping ability were analyzed with two- and three-way ROBUST ANOVA. The significance level was set at p = 0.05.

Results

There was a statistically significant difference between all groups for shaping time (p < 0.001). There was a statistically significant difference between groups for the total amount of material removed (p < 0.001). The directions of canal transportation were the inner surface of the curvature for W and TR and the outer for TE. There was no statistically significant difference between the groups for the amount of transportation (p > 0.05). The centering ratio of TE and W was statistically higher than TruNatomy (p < 0.001).

Conclusions

TruNatomy removed the least material that supported the minimally invasive endodontic approach. All file systems caused similar transportation and did not change the original canal shape.

Introduction

Root canal preparation is one of the most critical stages of root canal treatment. Root canal preparation aims to clean and shape the root canal system by creating a conical form that narrows from coronal to apical (Schilder, 1974).

Root canal preparation can be done with hand or endodontic motorized instruments. The insufficient flexibility of stainless steel instruments may cause problems in maintaining canal anatomy in highly curved canals (Alaçam, 2012).

To eliminate these drawbacks, tools made from nickel-titanium (Ni-Ti) alloys are currently used in root canal shaping techniques (Thompson & Dummer, 1997). Due to its superelasticity characteristic, it is thought that Ni-Ti based rotary instruments can preserve the original form of the canal without causing procedural errors such as ledge and perforation, especially in narrow and very curved root canals during their preparation (Esposito & Cunningham, 1995). Ni-Ti rotary files have developed throughout time and are now frequently utilized in clinical settings. The manufacturing process influences the mechanical characteristics of Ni-Ti tools in addition to the updated file generations.

Shaping ability is associated with achieving a continuous tapering canal shape, while centring ability refers to the ability of the axis of the file to be in-line with the axis of the canal during preparation and as such to cause no canal zipping, ledging or perforation (Vallaeys, Chevalier & Arbab-Chirani, 2016). To keep the original canal form centered, a variety of root canal shaping approaches have been proposed using various Ni-Ti systems (as M-wire, R-phase, traditional NiTi) and varied dynamics (continuous rotation, reciprocating motion, and adaptive motion) (Pedullà et al., 2016).

Rotary file systems have a higher rate of staying in the center of the canal compared to reciprocation-based systems. It is considered that the cause is progressive shaping. Clinical tests have revealed that the reciprocation movement minimizes ledge formation and transportation while also saving time through faster shaping (Zhao et al., 2014; Hamid et al., 2018; Marceliano-Alves et al., 2015; Wu, Fan & Wesselink, 2000).

When comparing reciprocal and rotary Ni-Ti files, some studies suggest that reciprocal Ni-Ti files remove more debris, while others state the reverse or find no difference. In studies where rotational movement was discovered to produce more debris than reciprocal movement, the discrepancy was attributed to the reciprocal system’s smaller file number. No file system has been shown to totally avoid apical debris extrusion (Silva et al., 2014; Tinaz et al., 2005; Ahn, Kim & Kim, 2016).

In reciprocation motion, the rotational direction changes on a regular basis. Counterclockwise movements are used for cutting, whereas clockwise movements relieve torsional stress on the file. As a result, the amplitude of the stress acting on the root dentin may diminish, as may the probability of root cracks and fractures.

According to studies, the occurrence of dentin defects after using different Ni-Ti tools varies between 4% and 80%. Four studies found no relationship between instrument kinematics and dentin microcrack formation, and no new cracks formed after instrumentation, whereas one study found that reciprocating files produced fewer microcracks in dentin than rotating files (Bürklein, Tsotsis & Schäfer, 2013; Nagendrababu & Ahmed, 2019).

Studies have shown that despite meticulous ultrasonic cleaning and decontamination of Ni-Ti rotary instruments, dental tissues and organic debris still adhere to surface cracks (Alapati et al., 2003; Alapati et al., 2004). Therefore, single-use of endodontic instruments is recommended to reduce instrument fatigue and possible cross-contamination. However, the single-use of endodontic instruments, especially the more expensive Ni-Ti rotary instruments, can become a financial burden for both endodontists and general dentists, particularly since current techniques involve the use of at least three to four Ni-Ti rotary instruments. Consequently, it would be beneficial to introduce canal preparation techniques that reduce the number of instruments necessary to achieve mechanical and biological objectives (Kenneth, Hargreaves & Cohen, 2016).

T-Endo MUST (TE) (Dentac, İstanbul, Türkiye) is the first domestically produced Ni-Ti root canal file working with the principle of reciprocal movement. The system consists of 4 files: glide path file TG (13/.04), shaping files M25 (25/.06), M40 (40/.04), and M50 (50/.04). TE is produced with a proprietary heat treatment called TM-Wire. It is claimed by the manufacturer that the files, which are produced with TM-wire and a unique heat treatment technology, are resistant to breakage and have the ability to be flexible even in the most curved channels.

It is used by turning 160° counterclockwise and 40° clockwise and recommended for used at 300 rpm and 4.0 torque. It provides minimally invasive shaping with 0.04 and 0.06 taper angles. T-Endo MUST files should be used with two mm movements up and down.

The glide path file (TG) and shaping files in the system have a square and “S” shaped cross-section, respectively. This cross-section allows for more debris removal from the walls and better cleaning of the ducts during duct forming (T-endo MUST, 2023).

The WaveOne Gold (W) file (Dentsply Sirona, Charlotte, NC, USA) which works with reciprocal motion, is produced from an alloy called Gold Wire, unlike the WaveOne file (Ruddle, 2016). After manufacture, a unique heating and slow cooling procedure generates gold wire files. Files that go through these processes become far more flexible and durable (Weber et al., 2015). The cross-section of the WaveOne Gold file is a parallelogram that is in contact with the canal walls at two points. Due to its structure which is not located in the center during movement, only two edges come into contact with the root canal walls. A space is created where debris can be transported coronally by reducing the contact point, and the screwing effect is reduced. The WaveOne Gold file works by moving 170° counterclockwise and 50° clockwise. It completes a full tour with a total of three reciprocal cycles (Ribeiro et al., 2023).

A new file system TruNatomy (TR) (Dentsply Sirona) files have been manufactured from thin 0.8 mm NiTi wire rather than the traditional 1.1 mm Ni-Ti wire used to fabricate most standard files. Also, a special heat treatment is applied during the manufacturing process.

The TR system consists of an orifice modifier, a glider with a centered cross-section parallelogram design, and shaping files available in three sizes, small (20/0.04 taper), prime (26/0.04 taper), and medium (36/0.03 taper) with an off-centered parallelogram cross sectional design. The manufacturer claims that TruNatomy files are more flexible and fatigue resistant because of their design (Vander Vyver, Vorster & Peters, 2019).

The primary objective of the study is to evaluate shaping ability, while the secondary objective is to identify procedural errors. The null hypothesis is that there are no differences between experimental files for shaping ability, apical transportation, and catering ratio.

Materials & Methods

Ethics committee approval was not required since the study was not conducted on human materials.

Selection of samples

In order to detect a statistically significant difference between the three groups (T-Endo Must, WaveOne Gold, TruNatomy) at an effect size of 0.4 with 80% power and α = 0.05 error level, the minimum sample size required to be included in the study is 22 resin blocks per group (66 resin blocks in total) has been calculated. G Power 3.1.9 in Calculation 2 package programs were used (Heinrich Heine, Düsseldorf, Germany).

Preparation of samples

Sixty-six resin blocks (VDW, Munich, Germany) with an apex diameter of 0.15 mm, a taper of 2%, a length of 19 mm, and a slope of 45° were equally divided into three groups according to file systems (Fig. 1). The simulated canals were all prepared in one session by the same endodontist using instruments up to the working length, which was established at canal terminus. The patency of all resin blocks was checked using K-file size 10 to eliminate any blockage with a manufacturing defect.

Figure 1 Unshaped resin block.

Initial imaging of resin blocks before shaping

In order to superimpose the digital photographs, a vertical and horizontal notch was prepared on the resin blocks prior to the acquisition of digital photographs. The resin blocks were painted with blue ink using a 30-gauge (G) side-hole irrigation needle (Navi-tip; Ultradent, UT, USA) before to localize the canal borders (Fig. 2). During shaping, resin blocks were covered with aluminum foil and fixed with a setup to prevent any influence on the researcher.

Figure 2 Resin block painted before shaping.

Creating sample groups and performing root canal preparations

Specimens were randomly divided into three groups (n = 22).

All preparations were carried out by an operator with three years of experience, using a single file system for two resin blocks. Shaping was performed using the X-Smart Plus (Dentsply, Tulsa Dental, Lancaster, PA, USA) endodontic motor. File systems were used according to the manufacturer’s recommendations. Each Ni-Ti file was used in two canals except for breakage or deformation. The time required for shaping root canals using the files was recorded with a chronometer in seconds, excluding the time for file change and irrigation.

Group TE: Shaped with T-Endo MUST Tg (13.04), T-Endo MUST M25 (25.06), and T-Endo MUST M40 (40.04) files.

Group W: Shaped with WaveOne Gold Small (20.07) and WaveOne Gold Medium (35.06) files.

Group TR: Shaped with TruNatomy Orifice Modifier (20.08), TruNatomy Glider (17.02), TruNatomy Small (20.04), TruNatomy Prime (26.04), and TruNatomy Medium (36.03) files (Fig. 3).

Figure 3 File systems used in the study.

During instrumentation, the root canals were constantly irrigated with 2 ml of distilled water, and 5 ml of distilled water was used for the final irrigation.

Considering the canal width of single rooted teeth, files with similar apical diameters within each system were selected as the final file to simulate clinical use.

Final imaging of resin blocks after shaping

The canals were photographed on a white background with a digital camera in a consistent, repeatable position with a fixed setting beneath the light source. The resin blocks were painted with red ink using a 30-gauge (G) side-hole irrigation needle after shaping to localize the canal borders (Fig. 4).

Figure 4 Resin block painted after shaping.

Evaluation of resin block samples

The pre- and post-instrumentation images were superimposed into a composite image by using a computer software program (Adobe Photoshop Inc, San Jose, CA, USA) with the help of guide grooves (Figs. 5, 6 and 7). Composite photos were loaded into the Rhinoceros 3D modeling software (Rhino 7; Robert McNeel and Associates for Windows, Washington, D.C., USA). The program was used for marking 22 spots (11 A-inner wall points and 11 B-outer wall points) on the superimposed pictures, that vary from apical to coronal. Measurement points were taken at one mm intervals; points 0 to 3 indicated the apical, points 4 to 7 the coronal curvature, and points 8 to 10 the coronal plane region of the canal (Figs. 8 and 9).

Figure 5 Superimposed images of groups.

Figure 6 Superimposed images of groups.

Figure 7 Superimposed images of groups.

Figure 8 Measurements in the superimposed image.

Figure 9 Measurements in the superimposed image.

Using the following equations (Ersev et al., 2010), these data provided a quantitative and qualitative evaluation of the shaping ability of TE, W and TR file systems.

The changes occurring in the original shape of the root canal were assessed using the following criteria:

(1) Total amount of material removal: The value obtained by combining the widths of resin removal from the 2 aspects of the canal.

(2) Amount of transportation: The absolute value of the difference between the widths of resin removal from the 2 aspects of the canal.

(3) Direction of transportation: Determined by the direction of the wider width of resin removal from the 2 aspects of the canal.

(4) Centering ratio: Calculated by dividing the narrower width of resin removal by the wider one from the 2 aspects of the canal.

Measurements were performed randomly on superimposed images, and the group was identified while recording the measurement results.

Also, canal aberrations were determined by using composite images. Assessments were performed based on the presence of an apical zip, narrowing, ledge, and danger zone. The canal aberrations were defined according to Ersev et al. (2010).

Statistical analysis

The WRS2 package in R and IBM SPSS V23 were used to analyze the data, and the Shapiro–Wilk Test was used to ensure conformity with normal distribution. The two-way ROBUST ANOVA test (using the median method) was used to investigate values that did not conform to normal distribution according to file and point, and the analysis results were provided as median (minimum-maximum). The three-way ROBUST ANOVA test (using the trimmed mean method) was used to investigate values that did not conform to normal distribution according to file, surface, and point, and the analysis results were provided as trimmed mean standard error. Multiple comparisons were performed using the Bonferroni correction.

According to the files, a one-way analysis of variance was used to compare the average times that followed a normal distribution, and multiple comparisons were made using the Tamhane test. The significance level was set at p < 0.05.

Results

Table 1 shows statistical differences between groups and regions by colour coding systems. In intergroup comparison, there was no statistically significant difference between groups for the amount of removed resin material from the root canal at the apical region (p > 0.001). When comparing the amount of resin removed from the coronal region and total root canal, there was a statistical difference between all groups (p < 0.001) (Table 2).

Table 1 Comparison of values by file, surface and regions.

	Q	p	
File	514.072	<0.001	
			
Regions	59.395	<0.001	
File*Surface	55.06	0.001	
File* Regions	15.326	0.007	
			
File*Surface* Regions	55.211	0.001	
Notes.

*Q: Three-Way ROBUST ANOVA Test Statistics.

Different indicators were employed in the statistical analyses. The significance levels for each of the bold, underlined, italic, and superscript symbols are specified in the table.

Table 2 Descriptive statistics of values by file, surface and regions.

Surface	File	Regions	Total	
		Apical	Middle	Coronal		
A	TE	0.177 ± 0.007 AB	0.148 ± 0.003 C	0.105 ± 0.001 DE	0.142 ± 0.005 A	
W	0.157 ± 0.006 AC	0.184 ± 0.004 B	0.112 ± 0.002 Eİ	0.150 ± 0.006 A	
TR	0.147 ± 0.005 AC	0.122 ± 0.004 İJ	0.087 ± 0.003 H	0.116 ± 0.005 C	
Total					
B	TE	0.062 ± 0.005 FG	0.099 ± 0.003 DH	0.112 ± 0.002 Eİ	0.097 ± 0.004 B	
W	0.101 ± 0.006 DEHİ	0.106 ± 0.004 DEİ	0.137 ± 0.004 CJ	0.117 ± 0.004 C	
TR	0.023 ± 0.004 K	0.049 ± 0.003 F	0.075 ± 0.002 G	0.052 ± 0.004 D	
Total					
Total(A+B)	TE	0.119 ± 0.014ABCDE	0.126 ± 0.006ABC	0.108 ± 0.002AD	0.117 ± 0.004c	
W	0.131 ± 0.007ABC	0.145 ± 0.009B	0.124 ± 0.003BC	0.131 ± 0.004b	
TR	0.083 ± 0.015ACDE	0.085 ± 0.008DE	0.081 ± 0.002E	0.082 ± 0.005a	
Total	0.116 ± 0.007ab	0.119 ± 0.004b	0.105 ± 0.002a	0.113 ± 0.002	
Notes.

* Trimmed means ± standard error.

a-c: There is no difference between main effects with the same letter; A-K: There is no difference between interactions with the same letter.

Each symbol given in the table can be compared within itself. Those written in bold can be compared statistically. Similarly, each group written in italics, underlined, double underlined, dashed underlined, wavy underlined and as superscript can be compared statistically within itself.

TE T-Endo Must file

W WaveOne Gold File

TR TruNatomy File

A inner surface of the canal wall

B outer surface of the canal wall

The direction and amount of canal transportation (mm) at all regions and the total root canal are shown in Tables 3 and 4, respectively.

Table 3 Descriptive statistics of Outside-Inside (B-A) values by file and regions.

File	Regions	Total	
	Apical	Middle	Coronal		
TE	0.09 (−0.1–0.24)A	−0.05 (−0.08–0.01)B	0.01 (−0.53–0.05)C	0.01 (−0.53–0.24)a	
W	−0.07 (−0.16–0.07)BD	−0.07 (−0.14–0.05)D	0.03 (−0.01–0.05)E	−0.06 (−0.16–0.07)b	
TR	−0.13 (−0.17–0.03)F	−0.08 (−0.26–0.02)D	−0.01 (−0.04–0.05)G	−0.07 (−0.26–0.05)c	
Total	−0.06 (−0.17–0.24)b	−0.06 (−0.26–0.02)b	0.01 (−0.53–0.05)a	−0.03 (−0.53–0.24)	
Notes.

* Median (minimum-maximum).

a-c: There is no difference between main effects with the same letter; A-G: There is no difference between interactions with the same letter.

TE T-Endo Must file

W WaveOne Gold File

TR TruNatomy File

A inner surface of the canal wall

B outer surface of the canal wall

Table 4 Descriptive statistics of absolute (B-A) values by file and regions.

File	Regions	Total	
	Apical	Middle	Coronal		
TE	0.1 (0.03–0.24)A	0.05 (0–0.08)B	0.01 (0–0.53)C	0.05 (0–0.53)	
W	0.07 (0–0.16)AB	0.07 (0.05–0.14)A	0.03 (0–0.05)D	0.06 (0–0.16)	
TR	0.13 (0.03–0.17)E	0.08 (0–0.26)AB	0.02 (0–0.05)CD	0.07 (0–0.26)	
Total	0.1 (0–0.24)c	0.06 (0–0.26)b	0.02 (0–0.53)a	0.06 (0–0.53)	
Notes.

* Median (minimum-maximum).

a-c: There is no difference between points with the same letter; A-E: There is no difference between interactions with the same letter.

TE T-Endo Must file

W WaveOne Gold File

TR TruNatomy File

A inner surface of the canal wall

B outer surface of the canal wall

In the apical region, TE tended toward the outer aspect of the curvature, whereas, W and TR tended toward the inner aspect of the curvature. In the total root canal, TE tended toward the outer aspect of the curvature, whereas W and TR tended toward the inner aspect of the curvature.

In the apical region, TE and W caused statistically significantly less amount of transportation than TR, and there was no statistical difference between TE and W groups (p > 0.001). In the total root canal, there was no statistical difference in the amount of transportation between groups (p > 0.001).

When the average rates of staying at the center in all three regions were compared between groups in Table 5, there was no statistically significant difference between the TE and W groups, but it was seen to be statistically significantly higher than the TR group (p < 0.001).

Table 5 Descriptive statistics of Less/More values by file and regions.

File	Regions	Total	
	Apical	Middle	Coronal		
TE	0.37 (0.22–0.86)A	0.69 (0.56–0.94)B	0.91 (0.38–0.97)C	0.69 (0.22–0.97)b	
W	0.63 (0.27–0.85)BD	0.62 (0.41–0.68)D	0.81 (0.67–0.92)E	0.67 (0.27–0.92)b	
TR	0.18 (−0.06–0.68)F	0.41 (0.18–0.77)A	0.82 (0.56–0.96)CE	0.44 (−0.06–0.96)a	
Total	0.38 (−0.06–0.86)c	0.62 (0.18–0.94)b	0.85 (0.38–0.97)a	0.65 (−0.06–0.97)	
Notes.

* Median (minimum–maximum).

a-c: There is no difference between main effects with the same letter; A-F: There is no difference between interactions with the same letter.

TE T-Endo Must file

W WaveOne Gold File

TR TruNatomy File

A inner surface of the canal wall

B outer surface of the canal wall

Less the side of the canal with less material removed

more the side of the canal with more material removed

Only two of the shaped samples exhibited danger zone formation. Both samples were parts of the W group. There were no additional shaping errors discovered. There was a statistically significant difference between all groups for shaping time (p < 0.001) (Table 6).

Table 6 Comparison of average times according to files (minutes).

	Mean ± sd	Median (min-max)	Test statistics	p *	
T-Endo MUST	1.6 ± 0.23c	1.6 (1.1–2.22)	51.047	<0.001	
TruNatomy	1.12 ± 0.1b	1.13 (0.95–1.3)	
WaveOne Gold	1.33 ± 0.1a	1.33 (1.2–1.52)	
Notes.

* a-c: There is no difference between groups with the same letter.

Discussion

The aim of the present study was to evaluate the changes in canal morphology produced by the file systems employed during canal shaping in resin blocks and to apply these findings in clinical settings.

There is a methodological gap in the study. The gap aries from the absence of extracted human teeth and the lack of a three-dimensional evaluation of shaping ability.

For statistical evaluations, median values were chosen by the statistician, as they provide the most accurate results for the selected tests.

Resin blocks are valuable since they standardize the canal’s size, form, and curvature while posing minimal risk of infection. Because resin blocks are transparent, instrumentation can be directly watched, allowing changes in canal shape to be detected more quickly than with dentin. Another advantage is that the artificial canal diameter and curvature are mathematically determined, which allows for direct comparison of various devices or instrumentation procedures. Thompson & Dummer (1998) employed clear resin blocks in various investigations to assess the effectiveness of Ni-Ti rotary instruments and concluded that these blocks are reliable.

There are two notable limitations associated with the use of resin blocks: Firstly, caution must be exercised when extrapolating the findings to clinical situations. Secondly, the heat generated by rotating instruments in resin blocks may soften the resin material, causing the cutting blade to bind and potentially leading to instrument separation. This occurs because the hardness of resin blocks is approximately half that of natural human dentine and they exhibit markedly different thermal properties (Al-Dhbaan et al., 2018; Shen & Cheung, 2013).

Clinicians experience a variety of challenges during root canal procedures, including anatomical variances, damaged root canals in retreatment situations, and limited mouth opening. J-shaped root canal forms are one of the issues that dentists frequently face. J-shaped root canals are widespread in clinical practice, with rates as high as 27% in maxillary canines and 24% in mandibular central incisors (Beshkenadze & Chipashvili, 2015). Old or novel generation procedures used to prepare curvature and narrow canals may induce iatrogenic complications such as step creation, canal obstruction, deformation of root canal morphology, transportation, and perforations (Webber, 2015).

Although several irrigation solutions such as NaOCl, EDTA, and CHX are recommended in clinical use, distilled water was chosen to avoid the potential effects of these solutions on the resin material, as pointed out in previous studies (Ünlü, Güneç & Haznedaroğlu, 2023; Burroughs et al., 2012).

In numerous studies, the shaping ability of files on resin blocks has been evaluated using ink staining (Özyürek, Yılmaz & Uslu, 2017; Alrahabi & Zafar, 2018; Almnea et al., 2024; Mahmoud et al., 2024). Ink staining makes the areas where the instruments have contacted and altered the canals visually distinct during the shaping process. It was considered that staining the canals with ink would not affect the results.

In a study evaluating shaping ability, rotary files have been used for more than one resin block (Mahmoud et al., 2024).

Digital images or micro-computed tomography (micro-CT) can be used to assess the shaping ability of Ni-Ti rotary files. Three-dimensional imaging technologies are useful for measuring the surface of materials and obtaining volumetric data. Ounsi et al. (2011) investigated the shaping efficiency of files using both two-dimensional photograph and three-dimensional microtomography approaches. 2D photograph or radiography procedures entail obtaining pictures and then running them through a software program to do calculations with or without overlaying them. These procedures have been utilized to assess preparation form, canal transport, residual dentin after shaping, and the cutting effectiveness of various devices (Javaheri & Javaheri, 2007; Iqbal et al., 2003).

According to the results of this study, the W group removed statistically significantly more resin material in the total root canal, while the TR group removed less resin material. Several studies compared the TR system to other file systems and found that it removes less material from the canal walls (Pit et al., 2020; Kim, Jeon & Seo, 2021; Morales et al., 2021). Based on the minimally invasive approach, the TR method seeks to retain more dentin in the pericervical area by using devices with decreasing taper angles for shaping (Silva et al., 2022). TR files are made with a smaller beginning wire (0.8 mm diameter) than conventional files (1.1 mm diameter) (Vorster, Van Der Vyver & Markou, 2023). Although both W and TE groups use reciprocal movement, the difference can be explained to the W group’s higher taper angle. A study was conducted to compare the shaping capacity of single file systems with varying taper angles. It showed that the taper of the files was the predicted factor for shaping capacity and that files with larger taper angles removed more resin material than files with smaller tapered angles (Saleh et al., 2015). These findings back up the findings of the current investigation.

The absence of a statistically significant difference in the total amount of material removed in the middle region between the W and TE groups can be attributed to the reciprocating motion of both files, as well as the constant taper of the TE file compared to the decreasing taper of the W file.

In the within-group comparison of the total amount of material removed, no statistically significant differences were observed between the regions in any of the file groups. The TE file system has a constant taper, while the W and TR file systems have a decreasing taper towards the coronal direction. The resin blocks were manufactured with a tapering shape that narrows towards the apical end, with the coronal portion being wider than the apical portion. During enlargement, the file systems performed shaping consistent with their respective tapers from apical to coronal. The similar amount of material removed in all three regions across all groups can be associated with these considerations.

Alfadley et al. (2020) compared the transportation properties of W and XP Endo Shaper on resin blocks and showed that both file systems with distinct designs and operation principles tended towards the inner surface in the middle area, comparable to the results of the present study. All file systems establish transportation towards the outer surface of the slope in the coronal area (Alfadley et al., 2020). The difference between studies might be related to differences in image capture, image assessment programs, and evaluation levels among investigations. A past study comparing the transportation direction of rotational and reciprocational files showed that both file systems tended towards the inner surface of the canal without considering of movement type of files (Shi et al., 2022). In several studies comparing transportation, all file groups tended towards the inner surface in the apical area regardless of different creation ways (Gu et al., 2017; Pacheco-Yanes et al., 2019). These results are compatible with the present study.

A past study showed that no difference between the two motions for apical transportation, (Wu et al., 2015) some studies showed that rotational movement induces less apical transportation (Gergi et al., 2015; Nabavizadeh et al., 2014). Many studies comparing apical transportation after rotational and reciprocal motions showed that reciprocal motion is preferable (Maia Filho et al., 2015; Tambe et al., 2014; Dhingra, Ruhal & Miglani, 2015; Saber, Nagy & Schäfer, 2015).

In this study, the TR system had the lowest transport value towards the outside wall, which is consistent with previous findings that the TR file shapes the canal while keeping its original morphology (Mahmoud et al., 2024; Kim, Jeon & Seo, 2021).

Wu, Fan & Wesselink (2000) found that apical transportation of more than 0.3 mm had a detrimental effect on root canal filling. It has been claimed that apical leaking occurs when transit exceeds 0.3 mm. None of the file systems had an apical transfer value greater than 0.3 mm in this study. As a result, the file systems utilized in the study appear to be clinically appropriate.

In this study, TE and W had a better centering ratio than TR. Several studies compared the centering ability of W with different rotational file systems and showed that W had better centering ability than compared file systems (Shi et al., 2022; Webber, 2015; Al-Labed, Layous & Alzoubi, 2022), similar to the present study. A previous study evaluated the centering rate of TR, W, and other file systems and demonstrated that W had a better apical centering ratio than TR (Unno et al., 2022).

In another study, the centering ability of ProTaper and Hero 642 files were compared on L- and S-shaped resin blocks and it was shown that Hero 642 instruments preserved the original canal shape better and had better centering properties due to their fixed conical design (Yang et al., 2006). In the present study, the TE had a higher centering ratio and this finding can be explained by the consistent taper angle of the TE, similar to the previous study.

In contrast to our findings, a study found that the TR file had the best apical centering ability. This variation could be attributed to the preference for s-shaped resin blocks and the measurement being performed in different programs (Mahmoud et al., 2024).

Therefore, the null hypothesis is that there are no differences between experimental files for shaping ability, apical transportation, and catering ratio was rejected.

Overpreparation of anatomically thin root canal walls weakens dentin of root, resulting in root fracture or strip perforations. The danger zone is proportional to the amount of dentin removed from the angled root region. In a research comparing the danger zone formation of W and other file systems, it was showed that the W had the highest danger zone generation potential (Yüksel et al., 2022).

In a J-shaped canals, the danger zone is the outer wall of the slope. Shaping errors were visually identified by the operator from superimposed images without knowing which group they belonged to.

The working time is affected by movement technique and number of files used, working design of the file system, and the clinician’s expertise (Hülsmann, Peters & Dummer, 2005; Lim & Webber, 1985). By reducing shaping time, specialists may spend more time activating sodium hypochlorite and improving cleansing and disinfection (Plotino et al., 2015). Similar to the present study, a previous study compared the shaping times of TR and other file systems, and cocnluded that the TR file system had the shortest shaping time (Pit et al., 2020). In the present study, TE had the longest working time, and this finding might be based on prefered multiple reciprocal files. TE is a recently developed file system that has barely entered clinical use. There are just a few research regarding these files in the literature. One of the studies evaluated Tg file of TE system with other glide path files (Falakaloğlu & Uygun, 2021). Another research used glide path files to evaluate the shaping ability of the TE with the W. In that study, no difference was found between groups in the average amount of resin removed from the inner and outer canal walls (Falakaloğlu & İriboz, 2022). This finding differs from the finding of the present study and this difference might be due to the apical diameter set at 0.25.

In a study, it was reported that the TruNatomy file system achieves the final shaping of the root canal in minimal time (Pit et al., 2020). It is considered that the TR file quickly moves due to its design while shaping the canal walls in accordance with the anatomy through a minimally invasive approach; however, further studies are needed on this subject.

To the best of our knowledge, no study to date has analyzed the shaping ability and time of three file systems together in the literature. As a result, it was not possible to directly compare the current study findings with previous studies.

One of the benefits of the study is that it compares three file systems that have not been previously compared for clinical use. Another benefit is that it measures eleven points on resin blocks and examines three regions in detail to evaluate their shaping ability.

The study has some limitations. The first involves the use of resin blocks rather than excised human teeth. The second is the use of files for multiple canals. The third reason is that there are insufficient research on newly developed and clinically available information, making comparisons impossible.

More research should be undertaken on TE, TR, and W files from extracted human teeth, as well as procedures that analyze all three dimensions, such as CBCT.

Conclusion

Within the limitations of this study, TR using a minimally invasive approach removed less amount of material from resin blocks and also showed the shortest shaping time. TE and W had a better centering ratio. All file systems had similar total transportation amounts.

Supplemental Information

Supplemental Information 1 Measurements obtained during the experiment

Supplemental Information 2 Data obtained as a result of statistical analysis

This study was presented as an oral presentation at the 15th International Turkish Endodontic Society Congress. The authors would like to thank Gazi University Academic Writing Application and Research Center for proofreading the article. This article was previously published as a thesis (Karaca Sakallı, 2023).

Additional Information and Declarations

Competing Interests

Author Contributions

Data Availability

The authors declare there are no competing interests.

Aybüke Karaca Sakallı conceived and designed the experiments, performed the experiments, authored or reviewed drafts of the article, and approved the final draft.

Mügem Aslı Ekici conceived and designed the experiments, authored or reviewed drafts of the article, and approved the final draft.

The following information was supplied regarding data availability:

Raw data can be found in the Supplemental Files.

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
