# Peer review of "Evaluation of shaping ability, apical transportation and centering ratio of T-Endo Must, WaveOne Gold, and Trunatomy in resin blocks"

_PeerJ, doi:10.7717/peerj.18088_

## Round 0.1 · original submission · Major Revisions

I have received the reviewers’ comments for your manuscript titled. After a thorough review of their feedback, I have make a decision that your paper needs Major Revisions. The revisions to the manuscript will need you to address all the comments raised.

Reviewer 1 ·

Basic reporting

• Avoid double negative sentences. Reframe using proper nouns/ verbs/ conjunctions. Kindly refer doi.org/10.1002/bes2.1258
• Recheck for any wordy terms or buzzwords. Reframe words using stronger verbs, and do not repeat third-person pronouns.
• Recheck for any clipping of words or amber stands. Add footnotes in the manuscript describing the Logographic characters.
• Recheck- Report the conclusion in one sentence using simple direct-to-point words with an active voice
• Kindly mention primary & secondary objectives with the conformity of objectives
• Research gap is been identified. Add a note on the type of research gap (Evidence Gap, Knowledge Gap, Practical -Knowledge Conflict Gap, Methodological Gap, Empirical Gap, Theoretical Gap, Population Gap).
• How much percentage of the research gap is achieved could have been mentioned in the discussion part.
• Limitations of the study can be mentioned
• References as per Vancouver style needs to be done
• Keywords: Keywords as per Mesh terms needed. Kindly search keywords on MeSH on Demand, The MeSH Browser & Direct Browsing of MeSH Hierarchy (Trees). Use appropriate keywords applying Boolean terms (Please use keywords as per MESH headings on Medline)

Experimental design

• Kindly consider rewriting the methodology section with proper conceptual design, including hypothesis testing and experimental paradigm.
• Does your study support any other study hypothesis? If so, connect study your hypothesis with the research question.
• Add a note on the hypothesis that can connect to the introduction of your study
• Add a note on implications and limitations with reasonable explanations and conclude the study.
• Kindly remodify the manuscript by applying the argument matrix framework & linear construction method for a systematic approach to drafting. https://doi.org/10.1007/978-981-10-7062-4_6

Validity of the findings

None

Reviewer 2 ·

Basic reporting

I would like to thank the authors for their work on a new niti file however the tables were not clear: the abbreviations need to be defined in the legend, in addition to the different letter comparisons. which letters compare columns and rows. It is also not clear what is the more/less values and the outside/inside values. please explain in the table legend and material and methods.

Another figure showing the superimposed canals for each group of files in a clear way would also be relevant.
Please consider title change: It is very vague.
In the intro:
“insufficient flexibility of stainless steel instruments may cause problems with fit in” please consider rephrasing as the term “fit” is not clear.
“tools made of nickel-titanium”: made from
“causing procedural errors such as steps” the preferred term is ledge.
More info in required about the 3 systems used. Elaborate more about what makes each system unique. It is important to mention the files' cross-section.

Discuss in the intro the recipro and continuous motion rotary files and the pros/cons of the two systems.
Mention the interest of exploring canal centering ability of files.
Phrase the aim in a more detailed manner stating the null hypothesis.
Please add more recent studies from the literature in both the discussion and intro.

Experimental design

IRB exception should be mentioned with IRB no.
Sample size calculation should be mentioned.
The canal curvature parameters should be described in detail.
Who performed the assessment for the shaping errors and how?
How many operators performed the experiment. And at what intervals? What was the operator experience.
There is a major issue with using wave one gold for more than one canal as it is a single use file according to manufacturer instructions. Please address this.

How could the painting of the canals with ink influence the results?

What is the gap in research regarding these systems? What is the interest of this research? it was briefly mentioned in the discussion, but it should have been explained in more detail in the introduction.

Validity of the findings

In the Discussion: in the discussion the term “tissue hardness” please rephrase.

In the results “Only two of the shaped samples exhibited danger zone formation” what does this mean in a J shaped canal?

why were the medians compared and not the means? and what does this imply for the results.

Please include the limitations of this study and the relevance to clinical practice.
Please discuss the findings in relation for the sizes/tapers of the files.


Please rephrase: TR using a minimally invasive approach removed less amount of material from resin blocks and also showed the shortest shaping time. This is not clear.

Additional comments

To address this article from an overall perceptive: The title did not capture interest and there is no mention why these particular niti files were chosen and why they were used until these large sizes 40/.04?? and how is their performance related to their design or material. The paper would benefit from detail. Picture comparing the three files would also be nice.

Reviewer 3 ·

Basic reporting

The article is well structured and written. the introduction is lacking some references
Line 43 please provide reference for TM wire
Line 51 please provide reference for TruNatomy File
Line 54, at the end of introduction please mention the knowledge gap and Null hypothesis

Experimental design

Methodology
Line 55 Start with your power analysis
Line 58 How did you standardize your digital image acquisition?
Line 59 mention the criteria of the resin blocks; Size, Angle and radius of curvature
Any Blinding during measurements?
Discussion
Provide more information on why distilled water was chosen as the irrigant?
Address any limitations of the current study, such as the use of resin blocks instead of human teeth, and how these limitations might affect the results.

Validity of the findings

Data have been provided; they are robust, statistically sound, & controlled.

---

## Round 0.2 · accepted · Accept

Congratulations and thank you for addressing the comments

Reviewer 1 ·

Basic reporting

No comment

Experimental design

No comment

Validity of the findings

No comment

Additional comments

Dear Authors,
The authors have addressed all the comments and suggestions reviewers gave, and the manuscript has dramatically improved. The manuscript can be accepted for publication in its current form. I want to congratulate the authors and wish them the best in their future endeavours.

Best regards and keep well